# A Direct Approach to Robust Deep Learning Using Adversarial Networks

**Huaxia Wang**
Department of Electrical and Computer Engineering
Stevens Institute of Technology
Hoboken, NJ 07030, USA
hwang38@stevens.edu

**Chun-Nam Yu**
Nokia Bell Labs
600 Mountain Avenue
Murray Hill, NJ 07974, USA
chun-nam.yu@nokia-bell-labs.com

## Abstract

Deep neural networks have been shown to perform well in many classical machine learning problems, especially in image classification tasks. However, researchers have found that neural networks can be easily fooled, and they are surprisingly sensitive to small perturbations imperceptible to humans. Carefully crafted input images (adversarial examples) can force a well-trained neural network to provide arbitrary outputs. Including adversarial examples during training is a popular defense mechanism against adversarial attacks. In this paper we propose a new defensive mechanism under the generative adversarial network (GAN) framework. We model the adversarial noise using a generative network, trained jointly with a classification discriminative network as a minimax game. We show empirically that our adversarial network approach works well against black box attacks, with performance on par with state-of-art methods such as ensemble adversarial training and adversarial training with projected gradient descent.

## 1 Introduction

Deep neural networks have been successfully applied to a variety of tasks, including image classification (Krizhevsky et al., 2012), speech recognition (Graves et al., 2013), and human-level playing of video games through deep reinforcement learning (Mnih et al., 2015). However, Szegedy et al. (2014) showed that convolutional neural networks (CNN) are extremely sensitive to carefully crafted small perturbations added to the input images. Since then, many adversarial examples generating methods have been proposed, including Jacobian based saliency map attack (JSMA) (Papernot et al., 2016a), projected gradient descent (PGD) attack (Madry et al., 2018), and C&W's attack (Carlini & Wagner, 2017). In general, there are two types of attack models: white box attack and black box attack. Attackers in white box attack model have complete knowledge of the target network, including network's architecture and parameters. Whereas in black box attacks, attackers only have partial or no information on the target network (Papernot et al., 2017).

Various defensive methods have been proposed to mitigate the effect of the adversarial examples. Adversarial training which augments the training set with adversarial examples shows good defensive performance in terms of white box attacks (Kurakin et al., 2017; Madry et al., 2018). Apart from adversarial training, there are many other defensive approaches including defensive distillation (Papernot et al., 2016b), using randomization at inference time (Xie et al., 2018), and thermometer encoding (Buckman et al., 2018), etc.

In this paper, we propose a defensive method based on generative adversarial network (GAN) (Goodfellow et al., 2014). Instead of using the generative network to generate samples that can fool the discriminative network as real data, we train the generative network to generate (additive) adversarial noise that can fool the discriminative network into misclassifying the input image. This allows flexible modeling of the adversarial noise by the generative network, which can take in the original image or a random vector or even the class label to create different types of noise. The discriminative networks used in our approach are just the usual neural networks designed for their specific classification tasks. The purpose of the discriminative network is to classify both clean and adversarial example with correct label, while the generative network aims to generate powerful perturbations

to fool the discriminative network. This approach is simple and it directly uses the minimax game concept employed by GAN. Our main contributions include:

- We show that our adversarial network approach can produce neural networks that are robust towards black box attacks. In the experiments they show similar, and in some cases better, performance when compared to state-of-art defense methods such as ensemble adversarial training (Tramèr et al., 2018) and adversarial training with projected gradient descent (Madry et al., 2018). To our best knowledge we are also the first to study the joint training of a generative attack network and a discriminative network.

- We study the effectiveness of different generative networks in attacking a trained discriminative network, and show that a variety of generative networks, including those taking in random noise or labels as inputs, can be effective in attacks. We also show that training against these generative networks can provide robustness against different attacks.

The rest of the paper is organized as follows. In Section 2, related works including multiple attack and defense methods are discussed. Section 3 presents our defensive method in details. Experimental results are shown in Section 4, with conclusions of the paper in Section 5.

## 2 RELATED WORKS

In this section, we briefly review the attack and defense methods in neural network training.

### 2.1 ATTACK MODEL

Given a neural network model $D_\theta$ parameterized by $\theta$ trained for classification, an input image $x \in \mathbb{R}^d$ and its label $y$, we want to find a small adversarial perturbation $\Delta x$ such that $x + \Delta x$ is not classified as $y$. The minimum norm solution $\Delta x$ can be described as:

$$\arg\min_{\Delta x} \|\Delta x\| \quad \text{s.t.} \quad \arg\max D_\theta(x + \Delta x) \neq y, \tag{1}$$

where $\arg\max D_\theta(x)$ gives the predicted class for input $x$. Szegedy et al. (2014) introduced the first method to generate adversarial examples by considering the following optimization problem,

$$\Delta x = \arg\min_z \lambda\|z\| + L(D_\theta(x + z), \hat{y}), \quad x + z \in [0, 1]^d \tag{2}$$

where $L$ is a distance function measuring the closeness of the output $D_\theta(x+z)$ with some target $\hat{y} \neq y$. The objective is minimized using box-constrained L-BFGS. Goodfellow et al. (2015) introduced the fast gradient sign method (FGS) to generate adversarial examples in one step, which can be represented as $\Delta x = \epsilon \cdot \text{sign}(\nabla_x l(D_\theta(x), y))$, where $l$ is the cross-entropy loss used in neural networks training. Madry et al. (2018) argues with strong evidence that projected gradient descent (PGD), which can be viewed as an iterative version of the fast gradient sign method, is the strongest attack using only first-order gradient information. Papernot et al. (2017) presented a Jacobian-based saliency-map attack (J-BSMA) model to generate adversarial examples by changing a small number of pixels. Moosavi-Dezfooli et al. (2017) showed that there exist a single/universal small image perturbation that fools all natural images. Papernot et al. (2017) introduced the first demonstration of black-box attacks against neural network classifiers. The adversary has no information about the architecture and parameters of the neural networks, and does not have access to large training dataset.

### 2.2 DEFENSE MODEL

In order to mitigate the effect of the generated adversarial examples, various defensive methods have been proposed. Papernot et al. (2016b) introduced distillation as a defense to adversarial examples. Lou et al. (2016) introduced a foveation-based mechanism to alleviate adversarial examples.

The idea of adversarial training was first proposed by Szegedy et al. (2014). The effect of adversarial examples can be reduced through explicitly training the model with both original and perturbed adversarial images. Adversarial training can be viewed as a minimax game,

$$\theta^* = \arg\min_\theta \mathbb{E}_{x,y} \max_{\Delta x} l(D_\theta(x + \Delta x), y). \tag{3}$$

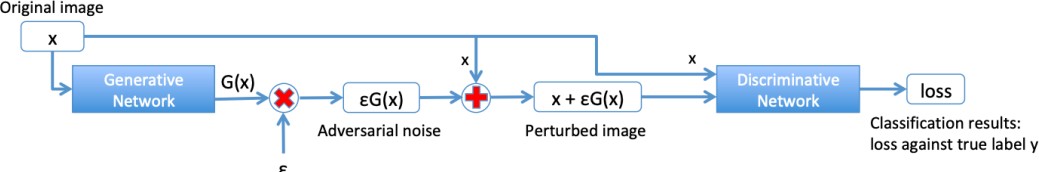

Figure 1: Architecture diagram of our adversarial networks

The inner maximization requires a separate oracle for generating the perturbations $\Delta x$. FGS is a common method for generating the adversarial perturbations $\Delta x$ due to its speed. Madry et al. (2018) advocates the use of PGD in generating adversarial examples. Moreover, a cascade adversarial training is presented in Na et al. (2018), which injects adversarial examples from an already defended network added with adversarial images from the network being trained.

There are a few recent works on using GANs for generating and defending against adversarial examples. Samangouei et al. (2018) and Ilyas et al. (2017) use GAN for defense by learning the manifold of input distribution with GAN, and then project any input examples onto this learned manifold before classification to filter out any potential adversarial noise. Our approach is more direct because we do not learn the input distribution and no input denoising is involved. Both Baluja & Fischer (2018) and Xiao et al. (2018) train neural networks to generate adversarial examples by maximizing the loss over a fixed pre-trained discriminative network. They show that they can train neural networks to effectively attack undefended discriminative networks while ensuring the generated adversarial examples look similar to the original samples. Our work is different from these because instead of having a fixed discriminative network, we co-train the discriminative network together with the adversarial generative network in a minimax game. Xiao et al. (2018) also train a second discriminative network as in typical GANs, but their discriminative network is used for ensuring the generated images look like the original samples, and not for classification. Lee et al. (2017) also considered the use of GAN to train robust discriminative networks. However, the inputs to their generative network is the gradient of the discriminative network with respect to the input image $x$, not just the image $x$ as in our current work. This causes complex dependence of the gradient of the generative network parameters to the discriminative network parameters, and makes the parameter updates for the generative network more complicated. Also there is no single minimax objective that they are solving for in their work; the update rules for the discriminative and generative networks optimize related but different objectives.

## 3 METHOD

In generative adversarial networks (GAN) (Goodfellow et al., 2014), the goal is to learn a generative neural network that can model a distribution of unlabeled training examples. The generative network transforms a random input vector into an output that is similar to the training examples, and there is a separate discriminative network that tries to distinguish the real training examples against samples generated by the generative network. The generative and discriminative networks are trained jointly with gradient descent, and at equilibrium we want the samples from the generative network to be indistinguishable from the real training data by the discriminative network, i.e., the discriminative network does no better than doing a random coin flip.

We adopt the GAN approach in generating adversarial noise for a discriminative model to train against. This approach has already been hinted at in Tramèr et al. (2018), but they decided to train against a static set of adversarial models instead of training against a generative noise network that can dynamically adapt in a truly GAN fashion. In this work we show that this idea can be carried out fruitfully to train robust discriminative neural networks.

Given an input $x$ with correct label $y$, from the viewpoint of the adversary we want to find additive noise $\Delta x$ such that $x + \Delta x$ will be incorrectly classified by the discriminative neural network to some other labels $\hat{y} \neq y$. We model this additive noise as $\epsilon G(x)$, where $G$ is a generative neural

network that generates instance specific noise based on the input $x$ and $\epsilon$ is the scaling factor that controls the size of the noise. Notice that unlike white box attack methods such as FGS or PGD, once trained $G$ does not need to know the parameters of the discriminative network that it is attacking. $G$ can also take in other inputs to generate adversarial noise, e.g., Gaussian random vector $z \in \mathbb{R}^d$ as in typical GAN, or even the class label $y$. For simplicity we assume $G$ takes in $x$ as input in the descriptions below.

Suppose we have a training set $\{(x_1, y_1), \ldots, (x_n, y_n)\}$ of image-label pairs. Let $D_\theta$ be the discriminator network (for classification) parameterized by $\theta$, and $G_\phi$ be the generator network parameterized by $\phi$. We want to solve the following minimax game between $D_\theta$ and $G_\phi$:

$$\min_\theta \max_\phi \sum_{i=1}^n l(D_\theta(x_i), y_i) + \lambda \sum_{i=1}^n l(D_\theta(x_i + \epsilon G_\phi(x_i)), y_i), \tag{4}$$

where $l$ is the cross-entropy loss, $\lambda$ is the trade-off parameter between minimizing the loss on normal examples versus minimizing the loss on the adversarial examples, and $\epsilon$ is the magnitude of the noise. See Figure 1 for an illustration of the model.

In this work we focus on perturbations based on $\ell_\infty$ norm. This can be achieved easily by adding a tanh layer as the final layer of the generator network $G_\phi$, which normalizes the output to the range of $[-1, 1]$. Perturbations based on $\ell_1$ or $\ell_2$ norms can be accommodated by having the appropriate normalization layers in the final layer of $G_\phi$.

We now explain the intuition of our approach. Ideally, we would like to find a solution $\theta$ that has small risk on clean examples

$$R(\theta) = \sum_{i=1}^n l(D_\theta(x_i), y_i), \tag{5}$$

and also small risk on the adversarial examples under maximum perturbation of size $\epsilon$

$$R_{adv}(\theta) = \sum_{i=1}^n \max_{\Delta x, \|\Delta x\| \leq \epsilon} l(D_\theta(x_i + \Delta x), y_i). \tag{6}$$

However, except for simple datasets like MNIST, there are usually fairly large differences between the solutions of $R(\theta)$ and solutions of $R_{adv}(\theta)$ under the same model class $D_\theta$ (Tsipras et al., 2018). Optimizing for the risk under white box attacks $R_{adv}(\theta)$ involves tradeoff on the risk on clean data $R(\theta)$. Note that $R_{adv}(\theta)$ represent the risk under white box attacks, since we are free to choose the perturbation $\Delta x$ with knowledge of $\theta$. This can be approximated using the powerful PGD attack.

Instead of allowing the perturbations $\Delta x$ to be completely free, we model the adversary as a neural network $G_\phi$ with finite capacity

$$R_G(\theta) = \max_\phi \sum_{i=1}^n l(D_\theta(x_i + \epsilon G_\phi(x_i)), y_i). \tag{7}$$

Here the adversarial noise $G_\phi(x_i)$ is not allowed to directly depend on the discriminative network parameters $\theta$. Also, the generative network parameter $\phi$ is shared across all examples, not computed per example like $\Delta x$. We believe this is closer to the situation of defending against black box attacks, when the adversary does not know the discriminator network parameters. However, we still want $G_\phi$ to be expressive enough to represent powerful attacks, so that $D_\theta$ has a good adversary to train against. Previous work (Xiao et al., 2018; Baluja & Fischer, 2018) show that there are powerful classes of $G_\phi$ that can attack trained classifiers $D_\theta$ effectively.

In traditional GANs we are most interested in the distributions learned by the generative network. The discriminative network is a helper that drives the training, but can be discarded afterwards. In our setting we are interested in both the discriminative network and the generative network. The generative network in our formulation can give us a powerful adversary for attacking, while the discriminative network can give us a robust classifier that can defend against adversarial noise.

### 3.1 STABILIZING THE GAN TRAINING

The stability and convergence of GAN training is still an area of active research (Mescheder et al., 2018). In this paper we adopt gradient regularization (Mescheder et al., 2017) to stabilize the gradient descent/ascent training. Denote the minimax objective in Eq. 4 as $F(\theta, \phi)$. With the generative

network parameter fixed at $\phi_k$, instead of minimizing the usual objective $F(\theta, \phi_k)$ to update $\theta$ for the discriminator network, we instead try to minimize the regularized objective

$$F(\theta, \phi_k) + \frac{\gamma}{2}\|\nabla_\phi F(\theta, \phi_k)\|^2, \tag{8}$$

where $\gamma$ is the regularization parameter for gradient regularization. Minimizing the gradient norm $\|\nabla_\phi F(\theta, \phi_k)\|^2$ jointly makes sure that the norm of the gradient for $\phi$ at $\phi_k$ does not grow when we update $\theta$ to reduce the objective $F(\theta, \phi_k)$. This is important because if the gradient norm $\|\nabla_\phi F(\theta, \phi_k)\|^2$ becomes large after an update of $\theta$, it is easy to update $\phi$ to make the objective large again, leading to zigzagging behaviour and slow convergence. Note that the gradient norm term is zero at a saddle point according to the first-order optimality conditions, so the regularizer does not change the set of solutions. With these we update $\theta$ using SGD with step size $\eta_D$:

$$\theta_{l+1} = \theta_l - \eta_D \nabla_\theta [F(\theta_l, \phi_k) + \frac{\gamma}{2}\|\nabla_\phi F(\theta_l, \phi_k)\|^2]$$
$$= \theta_l - \eta_D [\nabla_\theta F(\theta_l, \phi_k) + \gamma \nabla^2_{\theta\phi} F(\theta_l, \phi_k) \nabla_\phi F(\theta_l, \phi_k)]$$

The Hessian-vector product term $\nabla^2_{\theta\phi} F(\theta_l, \phi_k) \nabla_\phi F(\theta_l, \phi_k)$ can be computed with double back-propagation provided by packages like Tensorflow/PyTorch, but we find it faster to compute it with finite difference approximation. Recall that for a function $f(x)$ with gradient $g(x)$ and Hessian $H(x)$, the Hessian-vector product $H(x)v$ can be approximated by $(g(x + hv) - g(x))/h$ for small $h$ (Pearlmutter, 1994). Therefore we approximate:

$$\nabla^2_{\theta\phi} F(\theta_l, \phi_k) \nabla_\phi F(\theta_l, \phi_k) \approx \nabla_\theta [\frac{F(\theta_l, \phi_k + hv) - F(\theta_l, \phi_k)}{h}], \tag{9}$$

where $v = \nabla_\phi F(\theta_l, \phi_k)$. Note that $\phi_k + hv$ is exactly a gradient step for generative network $G_\phi$. Setting $h$ to be too small can lead to numerical instability. We therefore correlate $h$ with the gradient step size and set $h = \eta_G/10$ to capture the curvature at the scale of the gradient ascent algorithm.

We update the generative network parameters $\phi$ with using (stochastic) gradient ascent. With the discriminative network parameters fixed at $\theta_l$ and step size $\eta_G$, we update:

$$\phi_{k+1} = \phi_k + \eta_G \nabla_\phi F(\theta_l, \phi_k).$$

We do not add a gradient regularization term for $\phi$, since empirically we find that adding gradient regularization to $\theta$ is sufficient to stabilize the training.

## 3.2 GENERATIVE AND DISCRIMINATIVE NETWORK PARAMETER UPDATES

In the experiments we train both the discriminative network and generative network from scratch with random weight initializations. We do not need to pre-train the discriminative network with clean examples, or the generative network against some fixed discriminative networks, to arrive at good saddle point solutions.

In our experiments we find that the discriminative networks $D_\theta$ we use tend to overpower the generative network $G_\phi$ if we just perform simultaneous parameter updates to both networks. This can lead to saddle point solutions where it seems $G_\phi$ cannot be improved locally against $D_\theta$, but in reality can be made more powerful by just running more gradient steps on $\phi$. In other words we want the region around the saddle point solution to be relatively flat for $G_\phi$. To make the generative network more powerful so that the discriminative network has a good adversary to train against, we adopt the following strategy. For each update of $\theta$ for $D_\theta$, we perform multiple gradient steps on $\phi$ using the same mini-batch. This allows the generative network to learn to map the inputs in the mini-batch to adversarial noises with high loss directly, compared to running multiple gradient steps on different mini-batches. In the experiments we run 5 gradient steps on each mini-batch. We fix the tradeoff parameter $\lambda$ (Eq. 4) over loss on clean examples and adversarial loss at 1. We also fix the gradient regularization parameter $\gamma$ (Eq. 8) at 0.01, which works well for different datasets.

## 4 EXPERIMENTS

We implemented our adversarial network approach using Tensorflow(Abadi et al., 2016), with the experiments run on several machines each with 4 GTX1080 Ti GPUs. In addition to our adversarial

networks, we also train standard undefended models and models trained with adversarial training using PGD for comparison. For attacks we focus on the commonly used fast gradient sign (FGS) method, and the more powerful projected gradient descent (PGD) method.

For the fast gradient sign (FGS) attack, we compute the adversarial image by

$$\hat{x}_i = \text{Proj}_X \left( x_i + \epsilon \, \text{sign} \, \nabla_x l(D_\theta(x_i), y_i) \right), \tag{10}$$

where $\text{Proj}_X$ projects onto the feasible range of rescaled pixel values $X$ (e.g., [-1,1]).

For the projected gradient descent (PGD) attack, we iterate the fast gradient sign attack multiple times with projection, with random initialization near the starting point neighbourhood.

$$\hat{x}_i^0 = \text{Proj}_X \left( x_i + \epsilon u \right)$$
$$\hat{x}_i^{k+1} = \text{Proj}_{B_\epsilon^\infty(x_i) \cap X} \left( \hat{x}_i^k + \delta \, \text{sign} \, \nabla_x l(D_\theta(\hat{x}_i^k), y_i) \right), \tag{11}$$

where $u \in \mathbb{R}^d$ is a uniform random vector in $[-1, 1]^d$, $\delta$ is the step size, and $B_\epsilon^\infty(x_i)$ is an $\ell_\infty$ ball centered around the input $x_i$ with radius $\epsilon$. In the experiments we set $\delta$ to be a quarter of the perturbation $\epsilon$, i.e., $\epsilon/4$, and the number of PGD steps $k$ to be 10. We adopt exactly the same PGD attack procedure when generating adversarial examples in PGD adversarial training. Our implementation is available at `https://github.com/whxbergkamp/RobustDL_GAN`.

## 4.1 MNIST

For MNIST the inputs are black and white images of digits of size 28x28 with pixel values scaled between 0 and 1. We rescale the inputs to the range of [-1,1]. Following previous work (Kannan et al., 2018), we study perturbations of $\epsilon = 0.3$ (in the original scale of [0,1]). We use a simple convolutional neural network similar to LeNet5 as our discriminator networks for all training methods. For our adversarial approach we use an encoder-decoder network for the generator. See Model D1 and Model G0 in the Appendix for the details of these networks. We use SGD with learning rate of $\eta_D = 0.01$ and momentum 0.9, batch size of 64, and run for 200k iterations for all the discriminative networks. The learning rates are decreased by a factor of 10 after 100k iterations. We use SGD with a fixed learning rate $\eta_G = 0.01$ with momentum 0.9 for the generative network. We use weight decay of 1E-4 for standard and adversarial PGD training, and 1E-5 for our adversarial network approach (for both $D_\theta$ and $G_\phi$). For this dataset we find that we can improve the robustness of $D_\theta$ by running more updates on $G_\phi$, so we run 5 updates on $G_\phi$ (each update contains 5 gradient steps described in Section 3.2 ) for each update on $D_\theta$.

Table 1(left) shows the white box attack accuracies of different models, under perturbations of $\epsilon = 0.3$ for input pixel values between 0 and 1. Adversarial training with PGD performs best under white box attacks. Its accuracies stay above 90% under FGS and PGD attacks. Our adversarial network model performs much better than the undefended standard training model, but there is still a gap in accuracies compared to the PGD model. However, the PGD model has a small but noticeable drop in accuracy on clean examples compared to the standard model and adversarial network model.

Table 1(right) shows the black box attack accuracies of different models. We generate the black box attack images by running the FGS and PGD attacks on surrogate models A', B' and C'. These surrogate models are trained in the same way as their counterparts (standard - A, PGD - B, adversarial network - C) with the same network architecture, but using a different random seed. We notice that the black box attacks tend to be the most effective on models trained with the same method (A' on A, B' on B, and C' on C). Although adversarial PGD beats our adversarial network approach on white box attacks, they have comparable performance on these black box attacks. Interestingly, the adversarial examples from adversarial PGD (B') and adversarial networks (C') do not transfer well to the undefended standard model. The undefended model still have accuracies between 85-95%.

## 4.2 SVHN

For the Street View House Number(SVHN) data, we use the original training set, augmented with 80k randomly sampled images from the extra set as our training data. The test set remains the same and we do not perform any preprocessing on the images apart from scaling it to the range of [-1,1]. We study perturbations of size $\epsilon = 0.05$ (in the range of [0,1]). We use a version of ResNet-18(He

| training method\attack | white box | | | black box | | | | | |
|---|---|---|---|---|---|---|---|---|---|
| | No Noise | FGS | PGD | FGS(A') | PGD(A') | FGS(B') | PGD(B') | FGS(C') | PGD(C') |
| standard(A) | **99.40%** | 23.70% | 0.00% | 39.49% | 3.41% | 90.56% | 86.10% | 94.21% | 91.36% |
| adversarial PGD(B) | 98.70% | **95.46%** | **92.92%** | **95.78%** | **96.18%** | **95.58%** | **95.01%** | **97.05%** | **96.48%** |
| adversarial network(C) | **99.32%** | 94.66% | 87.09% | **95.75%** | **96.19%** | **96.15%** | **95.24%** | **96.96%** | **95.78%** |

Table 1: Classification accuracies under white box and black box attack on MNIST ($\epsilon = 0.3$)

| training method\attack | white box | | | black box | | | | | |
|---|---|---|---|---|---|---|---|---|---|
| | No Noise | FGS | PGD | FGS(A') | PGD(A') | FGS(B') | PGD(B') | FGS(C') | PGD(C') |
| standard(A) | **96.34%** | 64.64% | 3.69% | 69.47% | 49.92% | 56.46% | 44.25% | 89.71% | **83.02%** |
| adversarial PGD(B) | 87.45% | 55.94% | **42.96%** | 85.21% | 83.46% | 59.09% | 48.20% | 87.41% | **83.23%** |
| adversarial network(C) | **96.34%** | **91.51%** | 37.97% | **90.02%** | **88.04%** | **75.34%** | **57.52%** | **91.48%** | 81.68% |

Table 2: Classification accuracies under white box and black box attacks on SVHN ($\epsilon = 0.05$)

et al., 2016) adapted to 32x32 images as our discriminative networks. For the generator in our adversarial network we use an encoder-decoder network based on residual blocks from ResNet. See Model D2 and Model G1 in the Appendix for details. For the discriminative networks we use SGD with learning rate of $\eta_D = 0.01$ and momentum 0.9, batch size of 64, weight decay of 1E-4 and run for 100k iterations, and then decrease the learning rate to 0.001 and run for another 100k iterations. For the generative network we use SGD with a fixed learning rate of $\eta_G = 0.01$ and momentum 0.9, and use weight decay of 1E-4.

Table 2(left) shows the white box attack accuracies of the models. Adversarial PGD performs best against PGD attacks, but has lower accuracies on clean data and against FGS attacks, since it is difficult to optimize over all three objectives with finite network capacity. Our adversarial network approach has the best accuracies on clean data and against FGS attacks, and also improved accuracies against PGD over standard training.

Table 2(right) shows the black box attack accuracies of the models. As before A', B', C' are networks trained in the same ways as their counterparts, but with a different random seed. We can see that the adversarial network approach performs best across most attacks, except the PGD attack from its own copy C'. It is also interesting to note that for this dataset, adversarial examples generated from the adversarial PGD model B' have the strongest attack power across all models. In the other two datasets, adversarial examples generated from a model are usually most effective against their counterparts that are trained in the same way.

### 4.3 CIFAR10

For CIFAR10 we scale the 32x32 inputs to the range of [-1,1]. We also perform data augmentation by randomly padding and cropping the images by at most 4 pixels, and randomly flipping the images left to right. In this experiment we use the same discriminative and generative networks as in SVHN. We study perturbations of size $\epsilon = 8/256$. We train the discriminative networks with batch size of 64, and learning rate of $\eta_D = 0.1$ for 100k iterations, and decrease learning rate to 0.01 for another 100k iterations. We use Adam with learning rate $\eta_G = 0.002$, $\beta_1 = 0.5$, $\beta_2 = 0.999$ for the generative network. We use weight decay 1E-4 for standard training, and 1E-5 for adversarial PGD and our adversarial networks.

Table 3(left) shows the white box accuracies of different models under attack with $\epsilon = 8/256$. The PGD model has the best accuracy under PGD attack, but suffer a considerably lower accuracy on clean data and FGS attack. One reason for this is that it is difficult to balance between the objective of getting good accuracies on clean examples and good accuracies on very hard PGD attack adversarial examples with a discriminative network of limited capacity. Our adversarial model is able to keep up with the standard model in terms of accuracies on clean examples, and improve upon it on accuracies against FGS and PGD attacks.

Table 3(right) shows the black box attack accuracies of the models. Our adversarial network method works better than the other approaches in general, except for the PGD attack from the most similar model C'. The adversarial PGD model also works quite well except against its own closest model

| training method\attack | white box | | | black box | | | | | |
|---|---|---|---|---|---|---|---|---|---|
| | No Noise | FGS | PGD | FGS(A') | PGD(A') | FGS(B') | PGD(B') | FGS(C') | PGD(C') |
| standard(A) | **91.59%** | 57.31% | 1.32% | 67.99% | 22.88% | **77.13%** | **75.06%** | 73.34% | 55.34% |
| adversarial PGD(B) | 75.30% | 47.63% | 41.16% | 74.04% | 74.23% | 57.73% | 55.72% | 73.31% | **73.09%** |
| adversarial network(C) | **91.08%** | **72.81%** | **44.28%** | **81.74%** | **79.48%** | **77.23%** | 74.04% | **78.51%** | 66.74% |

Table 3: Classification accuracies under white box and black box attack on CIFAR10 ($\epsilon = 8/256$)

| training method\attack | white box | | | black box | | | | | |
|---|---|---|---|---|---|---|---|---|---|
| | No Noise | FGS | PGD | FGS(A') | PGD(A') | FGS(B') | PGD(B') | FGS(C') | PGD(C') |
| ensemble(MNIST,$\epsilon=0.3$) | 98.65% | 2.52% | 0.00% | 90.91% | 93.91% | 90.94% | 88.12% | 90.79% | 89.61% |
| adv. net(MNIST,$\epsilon=0.3$) | **99.03%** | **94.66%** | **87.09%** | **95.75%** | **96.19%** | **96.15%** | **95.24%** | **96.96%** | **95.78%** |
| ensemble(SVHN,$\epsilon=0.05$) | 95.30% | 79.16% | 2.74% | **95.32%** | **93.88%** | 67.97% | 54.81% | **95.60%** | **93.21%** |
| adv. net(SVHN,$\epsilon=0.05$) | **96.34%** | **91.51%** | **37.97%** | 90.02% | 88.04% | **75.34%** | **57.52%** | 91.48% | 81.68% |
| ensemble(CIFAR10,$\epsilon=\frac{8}{256}$) | 87.17% | 57.91% | 11.35% | **85.01%** | **85.19%** | 68.76% | 67.41% | **79.64%** | **70.98%** |
| adv. net(CIFAR10,$\epsilon=\frac{8}{256}$) | **91.08%** | **72.81%** | **44.28%** | 81.74% | 79.48% | **77.23%** | **74.04%** | 78.51% | 66.74% |

Table 4: Classification accuracies under white box and black box attacks on ensemble adversarial training and adversarial networks on different datasets

B', and offers the smallest drop in accuracies in general. But its overall results are not the best since it suffers from the disadvantage of having a lower baseline accuracy on clean examples.

We have also performed experiments on CIFAR10 using a wider version of ResNet (Zagoruyko & Komodakis, 2016) by multiplying the number of filters by 10 in each of the convolutional layers. These wider version of ResNets have higher accuracies, but the relative strengths of the methods are similar to those presented here. In addition we have experiments on CIFAR100, and the results are qualitatively similar to CIFAR10. All these results are presented in the Appendix due to space restrictions.

## 4.4 Comparing against Ensemble Adversarial Training

We also compare against a version of ensemble adversarial training (Tramèr et al., 2018) on the above 3 datasets. Ensemble adversarial training works by including adversarial examples generated from static pre-trained models to enlarge the training set, and then train a new model on top of it. The quality of solutions depends on the type of adversarial examples included. Here we construct adversarial examples by running FGS (Eq. 10) and PGD (Eq. 11) on an undefended model, i.e., FGS(A) and PGD(A) in the previous tables. Here for FGS we substitute the target label $y$ with the most likely class $\arg\max D_\theta(x)$ to avoid the problem of label leakage. Following Tramèr et al. (2018) we also include another attack using the least likely class:

$$\hat{x}_i = \text{Proj}_X \left( x_i - \epsilon \, \text{sign} \, \nabla_x l(D_\theta(x_i), y_{LL}) \right), \quad (12)$$

where $y_{LL} = \arg\min D_\theta(x_i)$ is the least likely class. We include all these adversarial examples together with the original clean data for training. We use the same perturbations $\epsilon$ as in the respective experiments above.

Table 4 shows the results comparing ensemble adversarial training (EAT) with our adversarial networks approach. On MNIST, adversarial networks is better on white box attacks and also better on all black box attacks using models trained with standard training(A'), adversarial PGD(B'), and our adversarial networks approach(C') with different random seeds. On SVHN and CIFAR10 adversarial networks is better on white box attacks, and both methods have wins and losses on the black box attacks, depending on the attacks used. In general adversarial networks seem to have better white box attack accuracies since they are trained dynamically with a varying adversary. The black box accuracies depend a lot on the dataset and the type of attacks used. There is no definitive conclusion on whether training against a static set of adversaries as in EAT or training against a dynamically adjusting adversary as in adversarial networks is a better approach against black box attacks. This is an interesting question requiring further research.

| Generator | Original (A) | standard(A') | adversarial PGD(B') | adversarial network(C') |
|---|---|---|---|---|
| original accuracy | 91.59% | 90.54% | 75.71% | 89.21% |
| autoencoder(8 filters) | 31.07% | 41.40% | 74.56% | 84.59% |
| autoencoder(64 filters) | **6.08%** | **14.74%** | 75.52% | 86.64% |
| random Gaussian | 45.27% | 58.68% | 75.02% | 87.33% |
| label | 11.17% | 23.29% | **74.78%** | **82.43%** |

Table 5: Attack performance of various generator networks against an undefended network in terms of test accuracies. First column is the accuracy on the discriminative model $D_\theta$ that the generative attacker $G_\phi$ is trained on (similar to white box attacks). The next three columns are the attack accuracies on other models by the learned $G_\phi$ (similar to black box attacks)

| training method\attack | white box | | | black box | | | |
|---|---|---|---|---|---|---|---|
| | No Noise | FGS | PGD | FGS(A') | PGD(A') | FGS(B') | PGD(B') |
| autoencoder(8 filters) | 88.70% | 67.28% | 33.56% | 78.94% | 74.15% | 70.70% | 68.54% |
| autoencoder(64 filters) | 89.10% | 67.05% | 33.38% | 79.56% | 74.40% | 70.52% | 68.66% |
| random Gaussian | 89.73% | 69.43% | 35.16% | 80.09% | 76.02% | 71.22% | 69.66% |
| label | 88.72% | 67.09% | 37.70% | 80.95% | 77.80% | 70.90% | 68.81% |

Table 6: Classification accuracies under white box and black box attacks on CIFAR10 for adversarial networks trained with different generative adversaries ($\epsilon = 8/256$)

## 4.5 EXAMINING THE GENERATIVE NETWORKS

We also did a more in-depth study on the generative network with CIFAR10. We want to understand how the capacity of the generative network affects the quality of saddle point solution, and also the power of the generative networks themselves as adversarial attack methods. First we study the ability of the generative networks to learn to attack a fixed undefended discriminative network. The architectures of the generative networks (G1, G2, G3) are described in the Appendix. Here we study a narrow (G1, $k = 8$) and a wide version (G1, $k = 64$) of autoencoder networks using the input images as inputs, and also decoder networks $G(z)$ using random Gaussian vectors $z \in \mathbb{R}^d$ (G2) or networks $G(y)$ using the labels $y$ (G3) as inputs. We run SGD for 200k iterations with step size 0.01 and momentum of 0.9, and use weight decay of 1E-5. We report test accuracies on the original discriminator after attacks.

From Table 5 the wide autoencoder is more powerful than the narrow autoencoder in attacking the undefended discriminator network across different models. As a white-box attack method, the wide autoencoder is close to PGD in terms of attack power (6.08% vs 1.32% in Table 3(left)) on the undefended model. As a black-box attack method on the undefended model A', it works even better than PGD (14.74% vs 22.88% in Table 3(right)). However, on defended models trained with PGD and our adversarial network approach the trained generator networks do not have much effect. PGD is especially robust with very small drops in accuracies against these attacks.

It is interesting that generator network $G(z)$ with random Gaussian $z$ as inputs and $G(y)$ with label as input works well against undefended models A and A', reducing the accuracies by more than 30%, even though they are not as effective as using the image as input. $G(z)$ is essentially a distribution of random adversarial noise that we add to the image without knowing the image or label. $G(y)$ is a generator network with many parameters, but after training it is essentially a set of 10 class conditional 32x32x3 filters. We have also performed similar experiments on attacking models trained with adversarial PGD and our adversarial networks using the above generative networks. The results are included in the Appendix due to space restrictions.

We also co-train these different generative networks with our discriminative network (D2) on CI-FAR10. The results are shown in Table 6. It is slightly surprising that they all produce very similar performance in terms of white box and black box attacks, even as they have different attack powers against undefended networks. The generative networks do have very similar decoder portions, and this could be a reason why they all converge to saddle points of similar quality.

## 4.6 DISCUSSIONS

In the experiments above we see that adversarial PGD training usually works best on white box attacks, but there is a tradeoff between accuracies on clean data against accuracies on adversarial examples due to finite model capacity. We can try to use models with larger capacity, but there is always a tradeoff between the two, especially for larger perturbations $\epsilon$. There are some recent works that indicate training for standard accuracy and training for adversarial accuracy (e.g., with PGD) are two fairly different problems (Schmidt et al., 2018; Tsipras et al., 2018). Examples generated from PGD are particularly difficult to train against. This makes adversarial PGD training disadvantaged in many black box attack situations, when compared with models trained with weaker adversaries, e.g., ensemble adversarial training and our adversarial networks method.

We have also observed in the experiments that for black box attacks, the most effective adversarial examples are usually those constructed from models trained using the same method but with different random seed. This suggests hiding the knowledge of the training method from the attacker could be an important factor in defending against black box attacks. Defending against black box attacks is closely related to the question of the transferability of adversarial examples. Although there are some previous works exploring this question (Liu et al., 2017), the underlying factors affecting transferability are still not well understood.

In our experimentation with the architectures of the discriminative and generative networks, the choice of architectures of $G_\phi$ does not seem to have a big effect on the quality of solution. The dynamics of training, such as the step size used and the number of iterations to run for each network during gradient descent/ascent, seem to have a bigger effect on the saddle point solution quality than the network architecture. It would be interesting to find classes of generative network architectures that lead to substantially different saddle points when trained against a particular discriminative network architecture. Also, recent works have shown that there are connected flat regions in the minima of neural network loss landscapes (Garipov et al., 2018; Draxler et al., 2018). We believe that the same might hold true for GANs, and it would be interesting to explore how the training dynamics can lead to different GAN solutions that might have different robustness properties.

Our approach can be extended with multiple discriminative networks playing against multiple generative networks. It can also be combined with ensemble adversarial training, where some adversarial examples come from static pre-trained models, while some other come from dynamically adjusting generative networks.

## 5 CONCLUSIONS

We have proposed an adversarial network approach to learning discriminative neural networks that are robust to adversarial noise, especially under black box attacks. For future work we are interested in extending the experiments to ImageNet, and exploring the choice of architectures of the discriminative and generative networks and their interaction.

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

# APPENDIX

## NETWORK ARCHITECTURES

Our networks are mostly based on ResNet. Figure 2 shows the residual block used in our networks. We denote a residual block with $k$ copies of $d \times d$ filters, with a stride of $s$ in the first convolution as residual-block($d$, $s$, $k$). A stride of 2 means the inputs are downsampled by a factor of 2. The notation conv2d($d$, $s$, $k$) refers to a convolutional layer with $k$ copies of $d \times d$ filters, convolved with stride $s$, and similarly for the deconvolution deconv2d($d$, $s$, $k$). The notations maxpool($s$) and avgpool($s$) denote max-pooling and average pooling operations with strides $s$. FC denotes a fully connected layer, while BN denotes a batch normalization layer.

Figure 3 shows the discriminative networks used in this paper. D1 is a simple convolutional neural network used in the MNIST experiments. D2 is the standard version of ResNet. Figure 4 shows the

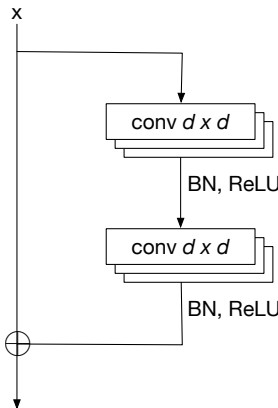

Figure 2: Residual block used in the network definitions

conv2d(5,1,32)
maxpool(2)
conv2d(5,1,64)
maxpool(2)
FC(10)

(a) D1

conv2d(3,1,16)
residual-block(3,1,16) ×3
residual-block(3,2,32)
residual-block(3,1,32) ×2
residual-block(3,2,64)
residual-block(3,1,64) ×2
avgpool(8)
FC(10)

(b) D2

Figure 3: Discriminative networks used in this paper

generative networks used in this paper. G0 and G1 are encoder-decoder networks, while G2 and G3 are decoder networks using a random vector and a one-hot encoding of the label respectively. The generative networks are parameterized by a factor $k$ determining the number of filters used (width of network). As default we use $k = 64$, and $k = 16$ for networks using labels as inputs.

EXTRA RESULTS ON CIFAR100 AND WIDE RESNET ON CIFAR10

The discriminative and generative networks in our CIFAR100 experiment have the same network architecture as the CIFAR10 experiment, except that the output layer dimension of the D network is 100 other than 10 in CIFAR10. We use learning rate of 0.1 for the first 100k iterations, and 0.01 for another 100k iterations. The batch size is 64 and weight decay is 1E-5.

Table 7(left) presents the white box attack accuracies of different models with $\epsilon = 8/256$. From the table we can see that the PGD adversarial training has the best defensive performance under PGD attack, but still suffers performance degradation on clean image and FGS attack. Our adversarial model gives similar classification performance as the standard model on clean image, and improves classification accuracies on FGS and PGD attack. Table 7(right) shows the black box attack accuracies of different models. Our adversarial network approach gives the best classification accuracy in most cases, except the FGS and PGD attack from model C'.

| training method\attack | white box | | | black box | | | | | |
|---|---|---|---|---|---|---|---|---|---|
| | No Noise | FGS | PGD | FGS(A') | PGD(A') | FGS(B') | PGD(B') | FGS(C') | PGD(C') |
| standard(A) | **70.11%** | 31.27% | 4.61% | 44.24% | 32.55% | **53.65%** | **50.51%** | 47.82% | 34.99% |
| adversarial PGD(B) | 55.53% | 32.13% | **28.48%** | 53.91% | 54.62% | 41.80% | 40.61% | **54.55%** | **53.57%** |
| adversarial network(C) | **70.99%** | **41.86%** | 18.25% | **58.11%** | **56.94%** | 53.15% | 51.87% | 53.35% | 46.61% |

Table 7: Classification accuracies under white box and black box attacks on CIFAR100 ($\epsilon = 8/256$)

Input: 28x28 image with $c$ channels
conv2d(5,2,64) - BN - ReLU
conv2d(5,2,128) - BN - ReLU
deconv2d(5,2,64) - BN - ReLU
deconv2d(5,2,$c$) - BN - ReLU
Tanh

(a) G0

Input: 32x32 image with $c$ channels
conv2d(3,1,$k$) - BN - ReLU
conv2d(3,2,$k$) - BN - ReLU
conv2d(3,2,2$k$) - BN - ReLU
residual-block(3,1,4$k$) ×6
deconv2d(3,2,2$k$) - BN - ReLU
deconv2d(3,2,$k$) - BN - ReLU
conv2d(3,1,$c$)
Tanh

(b) G1

Input: Random Gaussian $z \in \mathbb{R}^{256k}$
reshape(8,8,4$k$)
residual-block(3,1,4$k$) ×6
deconv2d(3,2,2$k$) - BN - ReLU
deconv2d(3,2,$k$) - BN - ReLU
conv2d(3,1,c)
Tanh

(c) G2

Input: One-hot encoding of label
FC(256$k$)
reshape(8,8,4$k$)
residual-block(3,1,4$k$) ×6
deconv2d(3,2,2$k$) - BN - ReLU
deconv2d(3,2,$k$) - BN - ReLU
conv2d(3,1,c)
Tanh

(d) G3

Figure 4: Generative networks used in this paper

| training method\attack | white box | | | black box | | | | | |
|---|---|---|---|---|---|---|---|---|---|
| | No Noise | FGS | PGD | FGS(A') | PGD(A') | FGS(B') | PGD(B') | FGS(C') | PGD(C') |
| standard(A) | **94.69%** | 62.36% | 1.08% | 69.89% | 9.14% | **85.05%** | **82.82%** | 76.27% | 45.16% |
| adversarial PGD(B) | 83.50% | 67.92% | **60.15%** | **83.21%** | **82.96%** | 72.66% | 68.82% | **82.01%** | **78.59%** |
| adversarial network(C) | 91.32% | **73.77%** | 49.55% | **83.29%** | **81.65%** | 79.32% | 76.00% | 79.32% | 62.71% |

Table 8: Classification accuracies under white box and black box attacks on CIFAR10 with Wide ResNet ($\epsilon = 8/256$)

Table 8 gives the results on CIFAR10 using a wider version of Resnet (Model D2), by multiplying the number of filters in each convolutional layer by a factor of 10. Some of the previous works in the literature use models of larger capacity for training adversarially robust models, so we perform experiments on these large capacity models here. First the accuracies increase across the board with larger capacity models. The accuracy gap on clean data between adversarial PGD and standard training still exists, but now there is also a small accuracy gap between our adversarial network approach and standard training. For the rest of the white box and black accuracies the story is similar, the models are weakest against attacks trained with the same method but with a different random seed. Our adversarial network approach has very good performance across different attacks, even as it is not always the winner for each individual attack. Table 9 gives the results of Wide ResNet on CIFAR100, and the results are qualitatively similar.

EXTRA RESULTS ON ATTACKING USING GENERATIVE NETWORKS AND THEIR TRANSFERABILITY

Following Section 4.5, we run extra experiments on using different generative networks to attack networks trained with adversarial PGD and our adversarial networks approach, in addition to the

| training method\attack | white box | | | black box | | | | | |
|---|---|---|---|---|---|---|---|---|---|
| | No Noise | FGS | PGD | FGS(A') | PGD(A') | FGS(B') | PGD(B') | FGS(C') | PGD(C') |
| standard(A) | **79.22%** | 44.86% | 6.38% | 48.52% | 13.42% | 65.18% | **63.56%** | 53.17% | 28.04% |
| adversarial PGD(B) | 66.68% | 45.54% | **38.36%** | **64.81%** | **64.84%** | 53.41% | 49.77% | **64.35%** | **62.44%** |
| adversarial network(C) | **80.21%** | **57.21%** | 30.27% | **65.37%** | 58.27% | **67.38%** | **64.28%** | 61.11% | 40.27% |

Table 9: Classification accuracies under white box and black box attacks on CIFAR100 with Wide Resnet ($\epsilon = 8/256$)

| Generator | Original (B) | standard(A') | adversarial PGD(B') | adversarial network(C') |
|---|---|---|---|---|
| original accuracy | 75.30% | 90.54% | 75.71% | 89.21% |
| autoencoder(8 filters) | 72.74% | 88.38% | 73.07% | 87.89% |
| autoencoder(64 filters) | 71.26% | 87.64% | 71.98% | 87.02% |
| random Gaussian | 74.13% | 88.71% | 74.70% | 88.45% |
| label | **70.15%** | **84.95%** | **70.10%** | **84.96%** |

Table 10: Attack performance of various generator networks against a network trained with adversarial PGD in terms of test accuracies. First column is the accuracy on the discriminative model $D_\theta$ that the generative attacker $G_\phi$ is trained on (similar to white box attacks). The next three columns are the attack accuracies on other models by the learned $G_\phi$ (similar to black box attacks)

| Generator | Original (C) | standard(A') | adversarial PGD(B') | adversarial network(C') |
|---|---|---|---|---|
| original accuracy | 91.08% | 90.54% | 75.71% | 89.21% |
| autoencoder(8 filters) | 74.66% | 66.95% | 74.42% | 80.55% |
| autoencoder(64 filters) | **53.68%** | **46.37%** | **73.08%** | **72.18%** |
| random Gaussian | 85.91% | 71.00% | 75.00% | 86.55% |
| label | 81.46% | 77.46% | **73.09%** | 83.98% |

Table 11: Attack performance of various generator networks against our adversarial network in terms of test accuracies. First column is the accuracy on the discriminative model $D_\theta$ that the generative attacker $G_\phi$ is trained on (similar to white box attacks). The next three columns are the attack accuracies on other models by the learned $G_\phi$ (similar to black box attacks)

undefended network in Section 4.5. Table 10 shows the results of various generative networks in attacking a network trained with adversarial PGD. The adversarial PGD network is very robust, and the generative networks can at most reduce the accuracy by 5%. Interestingly, the strongest attack come from the more restrictive generative network using only the label as input. It is also the most successful in transferring to other networks. However, since the adversarial PGD network is so robust, none of the generative networks can learn much from it in generating adversarial examples.

Table 11 shows the results of various generative networks in attacking our adversarial network. Our adversarial network is not as robust as adversarial PGD under white box attack, and the autoencoder(64 filters) network can reduce its accuracy from over 90% to 53%. Nonetheless, it is still much more robust than the undefended network. Interestingly, in addition to transferring well to the adversarial network trained with a different random seed (C'), the autoencoder(64 filters) network also transfers well to the undefended network, reducing its accuracy to 46%.

