# OpenReview forum: "A Direct Approach to Robust Deep Learning Using Adversarial Networks"
_ICLR.cc/2019/Conference_

### Official Review · AnonReviewer2 · 2018-11-02
**Robust defensive design using adversarial networks**

**Rating:** 6
**Confidence:** 3

**Review:**

The paper proposed a defensive mechanism against adversarial attacks using GANs. The general network structure is very much similar to a standard GANs -- generated perturbations are used as adversarial examples, and a discriminator is used to distinguish between them. The performance on MNIST, SVHN, and CIFAR10 demonstrate the effectiveness of the approach, and in general, the performance is on par with carefully crafted algorithms for such task.

pros
- the presentation of the approach is clean and easy-to-follow.
- the proposed network structure is simple, but it surprisingly works well in general.
- descriptions of training details are reasonable, and the experimental results across several datasets are extensive

cons
- the network structure may not be novel, though the performance is very nice.
- there are algorithms that are carefully crafted to perform the network defense mechanism, such as Samangouei et al, 2018. However, the method described in this paper, despite simple, works very good. It would be great if authors can provide more insights on why it works well (though not the best, but still reasonable), besides only demonstrating the experimental results.
- it would also be nice if authors can visualize the behavior of their design by showing some examples using the dataset they are working on, and provide side-to-side comparisons against other approaches.

---

> ### Author Response · Authors · 2018-11-23
> **Response to reviewer 2**
>
> Thank you for your valuable comments.
>
> For an explanation of why we think our adersarial network approach works well in training robust neural networks, we have provided more intuitions on our design in the Methods section on why it works well against black box attacks and also more observations in the Discussions section. These are by no means a definitive explanation, but more like a sharing of our intuitions and experiences with the models that hopefully can stimulate further discussions.
>
> As for the suggestions on visualizaiton, we have examined some of the perturbed images generated by our method and other methods. Qualitatively they don't look much different for the same perturbation size \epsilon, even if they have different losses. So we decide not to include such visualizations because they are not particularly information in this case. But during the revision of this paper we have expanded various sections and provided extra experiments in the appendix through the reviewers' feedbacks.

---

### Official Review · AnonReviewer3 · 2018-11-02
**An interesting method for robust deep learning**

**Rating:** 7
**Confidence:** 3

**Review:**

The paper "A Direct Approach to Robust Deep Learning Using Adversarial Networks" proposes a GAN solution for deep models of classification, faced to white and black box attacks. It defines an architecture where a generator network seeks to produce slight pertubations that succeed in fooling the discriminator. The discriminator is the targetted classification model.

The paper is globally well written and easy to follow. It well presents related works and the approach is well justified. Though the global idea is rather straightforward from my point of view, it looks to be a novel - effective - application of GANs. The implementation is well designed (it notably uses recent GAN stabilization techniques). The experiments are quite convincing, since it looks to produce rather robust models, without a loss of performance with clean (which appears crucial to me and  is not the case of its main competitors).

Minor comments:
    - eq1 : I do not understand the argmax (the support is missing). It corresponds to the class with higher probability I suppose but...
    - Authors say that GANs are usually useful for the generator (this is not always the case by the way), while in their case both obtained discriminator and generator have value. I do not understand in what the generator could be useful here, since it is only fitted to attack its own model (so what is the interest, are its attacks transferable on other models?)
    - Tables 1 and 2 are described as giving attack accuracies. But scores reported are classification accuracy right ? This is rather defense accuracies so...

---

> ### Author Response · Authors · 2018-11-23
> **Response to reviewer 3**
>
> Thank you for your valuable comments.
>
> Thank you for pointing out the ambiguity in the argmax. It is supposed to be referring to the maximum index in the last layer of the discriminative network (1,2,..., k), corresponding to the k classes. We have added a sentence to clarify this.
>
> For the usefulness of the generator, it is true that it learns to attack primarily the discriminative network it is trained against. However, the generator learned can sometimes be transferred to attack other models, although not necessarily as effective. In Table 5, the generator learned to attack model A, and were then used to attack other models A', B', and C'. It is very effective against A' trained using standard training but with a different random seed, somewhat effective against our adversarial network (C'), taking away close to 10% of accuracy. But it is not effective at all against adversarial PGD (B'). How well these learned generators can transfer to attack other models is an interesting question for further investigation.
>
> Thank you for pointing out the issues with table captions. We have replaced the captions with 'classification accuracies under white box and black box attacks' to avoid ambiguites.

---

### Official Review · AnonReviewer1 · 2018-11-05
**Propose the use of GANs to improve robustness to adversarial instances; extensive results but lack references and positioning to recent relevant arXiv papers**

**Rating:** 5
**Confidence:** 4

**Review:**

Summary: The paper proposes a GAN-based approach for dealing with adversarial instances, with the training of a robust discriminator that is able to identify adversaries from clean samples, and a generator that produces adversarial noise for its given input clean image in order to mislead the discriminator. In contrast to the state-of-the-art “ensemble adversarial training” approach, which relies on several pre-trained neural networks for generating adversarial examples, the authors introduce a way for dynamically generating adversarial examples on-the-fly by using a generator, which they along with their clean counterparts are then consumed for training the discriminator.

Quality: The paper is relatively well-written, although a little sketchy, and its motivations are clear. The authors compare their proposed approach with a good of variety of strong defenses such as “ensemble adversarial training” and “PGD adversarial training”, supporting with convincing experiments their approach.

Originality: Xioa et al. (2018) used very similar technique for generating new adversarial examples (generator attack), then used for training a robust discriminator. Likewise, Lee et al. (2018) also used GANs to produce perturbations for making images misclassified. Given this, what is the main novelty of this approach comparing to the (Xioa et al., 2018) and (Lee et al., 2018)? These references should be discussed in details in the paper.

Moreover, limited comparison with different attacks: Why did not compare against targeted attacks such as T-FGS, C&W or GAN-attack?

It is really surprising that undefended network is working better (showing more robustness) than the defended network “adversarial PGD” on black-box attacks, why this is happening?

References:
- Xiao, C., Li, B., Zhu, J. Y., He, W., Liu, M., & Song, D. (2018). Generating adversarial examples with adversarial networks. arXiv preprint arXiv:1801.02610.
- Lee, H., Han, S., & Lee, J. (2017). Generative Adversarial Trainer: Defense to Adversarial Perturbations with GAN. arXiv preprint arXiv:1705.03387.

---

> ### Author Response · Authors · 2018-11-23
> **Response to reviewer 1**
>
> Thank you for your valuable comments.
>
> We have already included Xiao et al. 2018 in the last paragraph of our related works section.
> Xiao et al. 2018 focused on using GAN as an attack method against a fixed pre-trained discriminative network, with another discriminative network co-trained to make sure the perturbed images look like the original. On the other hand our work is more focused on the defense using GAN, and we co-train the adversarial noise generator together with the discriminative network for classification (not the network for ensuring noisy images look like the original).  We didn't include Lee et al. 2018 in our original version, and it has been included in the updated veresion. Lee et al. 2017 also tried to use GAN to defend against adversarial attacks, but there are two main difference to our work. First, the inputs to their generative network is the gradient of the discriminative network with respect to the image x, not just the image x as in our current work. This causes complex dependence of the gradient of the generative network parameters to the discriminative network parameters, and makes the parameter updates for the generative network more complicated. Second, there doesn't seem to be a single minimax objective that they are solving for in their work; the update rules for networks D and G optimize related but different objectives. Our work on the other hand has only one minimax objective, and the updates on D and G networks are directly derived from it.
>
> And as for the attacks suggested by the reviewers, we didn't include them for various reasons.
> Targeted FGS is more useful when there are many close categories like ImageNet.  Since we are mostly dealing with datasets with small number of classes (10) in this paper, we believe the marginal benefit of including T-FGS is small since we already have FGS. For C&W attack, it is costly to run and Madry et al. 2018 showed that it is no more powerful than PGD attacks, so we stick to PGD attacks in this paper.  For GAN attacks, the results we have in Table 5 using generative networks to attack undefended models are equivalent to the GAN attacks used in earlier work by Xiao et al. 2018 and Baluja & Fischer 2017. We have also included extra results on GAN attacks on adversarial PGD and our adversarial networks in the Appendix.
>
> For the question on why undefended models sometimes work better than defended models under black box attacks, it really depends on how the black box attacks are constructed. What we have observed across experiments is that black box attack examples constructed (with FGS or PGD) from models trained using the same method but different random seeds are the most effective in attacking the same type of models. Therefore it is possible for the undefended model to work well against adversarial examples constructed from adversarial PGD(B') or our adversarial network(C') approach, as those adversarial examples might not transfer well to the undefended model. The undefended model is weak in the sense that it has low white box attack accuracies, and also it has the lowest black box accuracies when models of the same type (A') are used to attack it.

---

### Public Comment · (anonymous) · 2018-09-28
**Questions on Experimental Results**

I have a few concerns about the experiments on CIFAR10:
1. Your reported accuracy on clean data is relatively low.  In contrast, ResNet achieves an accuracy of 93%~95%. See, for example, https://github.com/bearpaw/pytorch-classification.
2. In Table 3, your method achieves ~75% accuracy against FGSM and PGD adversarial samples in the blackbox setting. However, I implement FGSM adversarial training to obtain ~85% accuracy under the exact same setting. FGSM is much simpler, while it yields better results. Am I missing anything?

---

> ### Author Response · Authors · 2018-10-16
> **Reply to questions on experimental results**
>
> 1. Thank you for pointing out this issue. We found that the weight decay we use (1E-5) was too small for CIFAR10. By changing the weight decay to 1E-4 the models can achieve accuracies of about 92% on clean data. We will update the results table accordingly. Using a deeper or wider network can push the results to 95% or above.
>
> 2. FGSM training, depending on how it's done, could lead to label leakage during test time. Also it's susceptible to the more powerful PGD attack. Due to space and training time constraints we just focus on adversarial training with the powerful PGD attack in this work.

---

> > ### Public Comment · (anonymous) · 2018-11-28
> > **Where are your updated results?**
> >
> > " We will update the results table accordingly. Using a deeper or wider network can push the results to 95% or above."
> >
> > I cannot find the updated results in the revised manuscript.

---

> > > ### Author Response · Authors · 2018-11-30
> > > **In the Appendix**
> > >
> > > The results for the wide resnet is in the Appendix due to space limitations.
> > >
> > > The tables in the paper itself are also updated after tuning of the weight decay parameter. Most of the numbers are close to the original.

---

### Public Comment · (anonymous) · 2018-10-09
**Prior defenses mentioned**

The paper writes prior defenses are "... defensive distillation Papernot et al. (2016b), using randomization at inference time Xie et al. (2018), and thermometer encoding (Buckman et al., 2018), etc." These might not be the best example to pick since these have been shown to be broken: https://arxiv.org/abs/1607.04311 and https://arxiv.org/abs/1802.00420

---

### Public Comment · (anonymous) · 2018-11-14
**No black-box query attacks were tried**

This paper makes several black-box claims but no attacks that query the model were tried (e.g., the Decision Attack from ICLR'18 or SPSA from Uesato et al. 2018 at ICML'18). Could the authors try either of these attacks?

---

### Meta-Review · Area_Chair1 · 2018-12-16
**A GAN approach to robust learning against adversarial examples.**

**Confidence:** 4
**Recommendation:** Accept (Poster)

**Metareview:**

The paper proposed a GAN approach to robust learning against adversarial examples, where a generator produces adversarial examples as perturbations and a discriminator is used to distinguish between adversarial and raw images. The performance on MNIST, SVHN, and CIFAR10 demonstrate the effectiveness of the approach, and in general, the performance is on par with carefully crafted algorithms for such task.

The architecture of GANs used in the paper is standard, yet the defensive performance seems good. The reviewers wonder the reason behind this good mechanism and the novelty compared with other works in similar spirits. In response, the authors add some insights on discussing the mechanism as well as comparisons with other works mentioned by the reviewers.

The reviewers all think that the paper presents a simple scheme for robust deep learning based on GANs, which shows its effectiveness in experiments. The understanding on why it works may need further explorations.  Thus the paper is proposed to be borderline lean accept.